# Locked Away—Prophylaxis and Management of Catheter Related Thrombosis in Hemodialysis

**DOI:** 10.3390/jcm10112230

**Published:** 2021-05-21

**Authors:** Joanna Szymańska, Katarzyna Kakareko, Alicja Rydzewska-Rosołowska, Irena Głowińska, Tomasz Hryszko

**Affiliations:** 2nd Department of Nephrology and Hypertension with Dialysis Unit, Medical University of Bialystok, 15-276 Białystok, Poland; katarzyna.kakareko@umb.edu.pl (K.K.); alicja.rosolowska@umb.edu.pl (A.R.-R.); irena.glowinska@umb.edu.pl (I.G.); tomasz.hryszko@umb.edu.pl (T.H.)

**Keywords:** central venous catheters, hemodialysis, locking solutions, catheter dysfunction, thrombosis, permanent catheter, tunneled catheter

## Abstract

Reliable vascular access is necessary for effective hemodialysis. Guidelines recommend chronic hemodialysis via an arteriovenous fistula (AVF), however, in a significant number of patients, permanent central venous catheters (CVCs) are used. The use of a tunneled catheter is acceptable if the estimated dialysis time is less than a year or it is not possible to create an AVF. The main complications associated with CVC include thrombosis and catheter-related bloodstream infections (CRBSIs), which may result in loss of vascular access. The common practice is to use locking solutions to maintain catheter patency and minimize the risk of CRBSI. This paperwork summarizes information on currently available locking solutions for dialysis catheters along with their effectiveness in preventing thrombotic and infectious complications and describes methods of dealing with catheter dysfunction. The PubMed database was systematically searched for articles about locking solutions used in permanent CVCs in hemodialysis patients. Additional studies were identified by searching bibliographies and international guidelines. Articles on end-stage kidney disease patients dialyzed through a permanent CVC were included. Information from each primary study was extracted using pre-determined criteria including thrombotic and infectious complications of CVC use, focusing on permanent CVC if sufficient data were available. Of the currently available substances, it seems that citrate at a concentration of 4% has the best cost-effectiveness and safety profile, which is reflected in the international guidelines. Recent studies suggest the advantage of 2+1 protocols, i.e., taurolidine-based solutions with addition of urokinase once a week, although it needs to be confirmed by further research. Regardless of the type of locking solution, if prophylaxis with a thrombolytic agent is chosen, it should be started from the very beginning to reduce the risk of thrombotic complications. In case of CVC dysfunction, irrespective of the thrombolysis attempt, catheter replacement should be planned as soon as possible.

## 1. Introduction

Reliable, properly functioning vascular access is key to effective hemodialysis and through this, it helps to ensure the best possible quality of life for the patient with end-stage kidney disease. In chronic maintenance hemodialysis patients, there are three options—arteriovenous fistula (AVF), arteriovenous graft (AVG) and tunneled central venous catheter (CVC). Among these, the access recommended by Kidney Disease Outcomes Quality Initiative (KDOQI) Guidelines as having the lowest complication rate is AVF [1]. The use of a tunneled catheter is acceptable if the estimated duration of dialysis treatment is less than a year, it is not possible to create an AVF or it is a valid patient preference [1]. In recent years in Poland, the percentage of patients with a permanent hemodialysis catheter has been increasing—in 2017, CVCs constituted 31% of created vascular access [2]. The frequency of CVC usage differs among countries; in 2013 in Canada, 45% of patients used CVCs, whereas it was 15% in the USA and less than 10% in Japan and Russia [3]. Due to their widespread use, every effort should be made to minimize the number of complications and ensure the longest possible period of functioning. The most common complications associated with central catheters include intra- or pericatheter thrombosis (catheter related thrombosis—CRT) and catheter-related bloodstream infections (CRBSI) [4]. Malfunction and CRT are responsible for 30–40% of cases of loss of vascular access [5].

There is no standard definition of catheter dysfunction. KDOQI defined CVC malfunction as failure to maintain the prescribed extracorporeal blood flow required for adequate hemodialysis without lengthening the prescribed HD treatment. It is noteworthy that the previous definition from the 2006 guidelines was more specific and defined it as an inability to achieve a minimum blood flow of 300 mL/min with a pre-pump arterial pressure below—250 mm Hg.

The inability to maintain the prescribed blood flow values leads to inadequate hemodialysis, which may result in worsening the patient’s quality of life, prognosis, and other negative health consequences. Thrombotic complications may require catheter removal or replacement, which increases the frequency of hospitalizations and generates a higher cost of patient care. Clots and fibrin deposits also promote biofilm formation and may increase the risk of another life-threatening complication—sepsis [6,7].

For many years, heparin has been the most commonly used locking solution. Recently, numerous studies have been conducted to assess the effectiveness of locking solutions available on the market, including citrate, urokinase, alteplase, taurolidine, antibiotics, used alone or in combination. The following paper summarizes information about the currently available locking solutions for dialysis catheters along with their effectiveness in preventing thrombotic and infectious complications and describes methods of dealing with catheter dysfunction.

## 2. Materials and Methods

Initially, the PubMed database was searched for systematic reviews and meta-analyses on locking solutions that were published in English. Reviews were identified using the search term “locking solution” and the selection included those that were conducted using a systematic search process and related to a population of chronic hemodialysis adult patients. No publication date or publication status restrictions were imposed. Previous versions of meta-analyses written by the same authors have been excluded to avoid duplication of data. In addition, hand-searching of references in selected reviews was performed. Surprisingly, no meta-analysis evaluating the effect of locking solutions exclusively in the population of patients chronically hemodialysis through permanent CVCs was found. Moreover, some studies have focused predominantly on CRBSI [8,9,10] or CRT [6], which may be a weak point, as both complications often coexist or pave the way for each other. Key systematic and comprehensive reviews and meta-analyses are summarized in Table 1. Systematic reviews and meta-analyses were assessed using the Preferred Reporting Items for Systematic Reviews and Meta-Analyses (PRISMA) checklist (http://www.prisma-statement.org/ accessed on 10 May 2021). For meta-analyses, the PRISMA checklist contained 24 required reporting items and three optional items (item 16 (description of additional analyses), item 19 (reporting of data on risk of bias for each study), and item 23 (reporting of results of additional analysis)). Only the required items were used for scoring. For systematic reviews, 19 items remained after the exclusion of optional items and items specific to meta-analyses (i.e., items 13, 14, 15, 21, and 22, which are related to data analysis and risk bias assessment).

In order to represent the current state of the literature, after identifying systematic reviews and meta-analyses, we also performed a systematic search for primary studies that were not included in the above-cited publications. Individual searches for clinical intervention trials of each locking solution was made. Used terms was, for heparin—“(heparin AND catheter AND hemodialysis)”; for trisodium citrate—“(citrate AND catheter AND hemodialysis)”; for taurolidine—“(taurolidine AND catheter AND hemodialysis)”; for alteplase and urokinase—“(alteplase OR urokinase AND catheter AND hemodialysis)”; for sodium bicarbonate—“(sodium bicarbonate AND catheter AND hemodialysis)”. The last search was run in November 2020. Previously described systematic reviews and meta-analyses were excluded to avoid data duplication. A database search was performed independently by two authors (J.S., T.H.). Subsequently, one review author (J.S.) extracted the following data from included studies and the second author (T.H.) checked the extracted data. Any disagreements were resolved by discussion between the authors. 

Clinical trials that were included in systematic reviews are reported in detail only if they provided key information. Due to limited data, studies were included regardless of the time of follow-up. The following categories of articles were exclude—wrong topics, editorials, reviews, practice guidelines, patient education materials, pediatric studies and non-English studies. Information from each primary study was extracted using pre-determined criteria including thrombotic and infectious complications of permanent CVC use.

A flowchart documenting the search strategy and results is shown in Figure 1.

## 3. Heparin

Since the first half of the 20th century, anticoagulant properties of heparin occurring through the subsequent inactivation of factor Xa and inhibition of the conversion of prothrombin into thrombin [14] have been widely used in medicine. Moreover, heparin is the longest-used locking solution to maintain catheter patency. For this reason, many researchers took it as a reference standard for assessing the effectiveness of other locking solutions. However, so far, the optimal concentration, which ranges from 1000 to 10.000 U/mL has not been clearly defined [11,15]. The use of concentrations of 5000 U/mL and higher is associated with an increased risk of bleeding complications, especially in the first period after tunneled catheter insertion [16]. Conversely, the use of low concentrations, such as 1000 U/mL had no detectable difference in time to malfunction compared to 5000 U/mL [17], however the study showed a 32% rate of catheter loss within 90 days. Despite numerous studies, it has not been clearly proven that any other substance administered into the lumen of the catheter in the interdialytic period has a higher efficacy in maintaining its patency [6,11].

Any substance administered into the lumen of the catheter, even in an amount based on the capped luminal volume, partially leaks into the circulation [14,16,18,19], therefore the potential systemic effects, especially the increased risk of bleeding, should always be taken into consideration. The side effects of heparin also include heparin thrombocytopenia, allergic reactions and osteoporosis [7,11]. Moreover, heparin does not have antibacterial properties. On the contrary, its presence has been shown to stimulate Staphylococcus aureus biofilm formation [20]. This bacteria is responsible for 35% of CRBSI [21]. The incidence of infectious complications during the use of dialysis catheters and the potential adverse effects are important features of catheter locks and should be taken into account when performing selection.

## 4. Trisodium Citrate

Citrate exerts an anticoagulant effect by chelating ionized calcium in the blood, thereby blocking the calcium-dependent components of the coagulation system as well as preventing platelet activation. The same mechanism of action is thought to be responsible for the reduction of biofilm formation, thus reducing the risk of inflammatory complications, and at the same time it does not cause bacterial resistance [20,21,22,23]. In vitro studies have also shown bactericidal and fungicidal properties of hypertonic citrate solutions. As in the case of heparin, different concentrations of the trisodium citrate solution were tested to determine which of them had the strongest anticoagulant and antibacterial effect while maintaining an appropriate safety profile. The most frequently tested solutions were 4%, 30% and 46.7%.

Wang et al., on the basis of their meta-analysis, found that the use of higher citrate concentrations does not reduce the incidence of thrombotic complications, moreover leads to a reduction in its antibacterial properties [11]. This is probably due to the precipitation of proteins inside the catheter and greater leakage of the lock solution into the patient’s circulation in the case of hypertonic solutions with a concentration higher than 12% [21,24,25]. As a consequence, the active substance inside the catheter lumen is replaced by the patient’s blood and its protective effect disappears [24]. Sodium citrate concentrations <10% are equally effective as heparin for preventing CVC malfunction and seem to have the best risk–benefit ratio [14,26,27].

Compared to heparin, the use of citrate is associated with a lower risk of major bleeding [14,27,28] and lower cost [14,27,29]. The most frequently reported adverse reaction was a metallic taste in the mouth [11,21]. Nausea, paresthesias in the face and fingers were reported at higher concentrations [11,25]. In 2000, the Food and Drug Administration (FDA) advised to avoid the use of 46.7% citrate due to the increased risk of a major hypocalcemia resulting in arrhythmias and sudden cardiac arrest.

The advantage of using high concentrations of citrate as a lock solution for dialysis catheters has not been proven. On the contrary, based on the studies carried out so far, it seems that the best balance of benefits against the risk of side effects is shown by solutions <10% [1,11,14,21,25,26,30], which is reflected in the KDOQI Guidelines suggesting the use of low-concentration citrate (<5%) CVC locking solution. In Europe, from a practical point of view, citrate at a concentration of 4% is available on the market.

## 5. Taurolidine

In vitro studies have shown that taurolidine has antibacterial and antifungal activity, and its use reduces the development of biofilm on vascular catheters [31]. Unlike antibiotics, it poses no risk of resistance [32]. However, by itself, it does not have anticoagulant properties, so it should be used in combination with another substance, most often with citrate and heparin [32].

The LOCK-IT-100 study did not show an advantage of the combined preparation of taurolidine with 3.5% citrate and 1000 IU/mL heparin over heparin alone in terms of loss of patency of the catheter or the need to remove it, but only in reducing the frequency of infection. Other studies have shown that the addition of a combined preparation of urokinase, used once a week, reduces the incidence of catheter dysfunction [33,34], but these studies were conducted in small groups of patients. In a randomized study, Winnicki et al. showed that the use of a solution of taurolidine, citrate and urokinase once a week, and taurolidine, citrate and 500 IU of heparin twice a week reduces the incidence of catheter dysfunction, the need for thrombolysis and the total cost of treatment compared to the control group in which only 4% citrate was used. These are promising results, but it is puzzling that episodes of dysfunction were relatively frequent, 18.7/1000 days in the test group and 44.3/1000 days in the control group, despite adopting criteria for blood flow values lower than indicated by the KDOQI. Moreover, in view of the research constructed in this way, it is difficult to determine whether the use of a mixture with taurolidine is of greater benefit, or only adding urokinase once a week, as suggested by the authors of the study [33]. Thus, in the future, more randomized controlled trials on a sufficiently large group are needed to confirm these results.

Adverse reactions associated with the use of lock solutions with taurolidine include dysgeusia [32,35], perioral dysesthesia, nausea, vomiting, and discomfort in the chest and neck [32].

## 6. Alteplase and Urokinase

Thrombolytic agents such as recombinant tissue plasminogen activator (rtPA, altepalse) and urokinase are used i.a. to restore the patency of a hemodialysis catheter. Their action is based on direct activation of plasminogen contained in the thrombotic material. Attempts have been made to verify whether these solutions will be effective in prophylaxis of catheter thrombosis and whether it will be financially advantageous compared to heparin or citrate. The results of the studies conducted so far indicate that the use of alteplase to fill the lumen of the catheter once a week reduces the frequency of dysfunction [11,36,37,38] but at a slightly higher cost compared to heparin [39] and at a very high total cost compared to 4% of citrate [36].

## 7. Sodium Bicarbonate

A potentially interesting alternative to the known and widely used anticoagulants discussed above is 8.4% sodium bicarbonate, which is rather commonly used for management of deep metabolic acidosis rather than filling of catheters. The mechanism of the anticoagulant action of NaHCO3 is similar to that of citrate—it chelates ionized calcium in the blood. Moreover, sodium bicarbonate has been shown to inhibit bacterial adhesion and biofilm formation [40] and, by changing the ion charge, it affects the structure of the bacterial membrane and gene expression, which stimulates the immune response [40]. A recently published study [40] comparing 8.4% NaHCO3 with 0.9% NaCl as locking solutions showed its significant effectiveness in preventing both thrombotic and infectious complications. No side effects were reported. The results are very interesting and potentially promising, especially considering the low cost of the substances. However, more research is needed to confirm the effectiveness of sodium bicarbonate [7].

The anticoagulant and antimicrobial properties of the locking solutions mentioned above are summarized in Figure 2. A summary of the estimated costs of locking solutions is provided in Table 2.

## 8. Management of Catheter Dysfunction

To receive an adequate blood flow through the dialyzer and perform an effective hemodialysis, a properly functioning, patent vascular access is necessary. The risk of catheter dysfunction is estimated at 15% within the first year after catheter insertion [41]. The causes can be classified as “early”, occurring up to 2 weeks after insertion of the catheter, and “late” [28,42]. Early dysfunction is often noticeable during and immediately after CVC insertion, usually due to technical issues in CVC placement, such as incorrect catheter tip placement, too tight suture, kinking or a manufacturing defect [28,42,43]. Early dysfunction should be diagnosed during or at the end of the insertion procedure and fixed immediately. A bigger problem is late malfunction, typically resulting from thrombus formation. The thrombotic material may occur inside the CVC lumen, at its tip or around the catheter, incorporated within a fibrin sheath, involving even the entire vena cava as mural thrombus [42].

Initial management in case of catheter malfunction is the same, regardless of the possible cause. First, any obvious mechanical obstruction should be excluded by carefully inspecting the catheter along its tunnel under the skin [44]. Then, non-invasive methods are tried, although their effectiveness is not fully verified. Patient repositioning (raising the ipsilateral arm, having the patient sit or stand, deep breathing or rolling from side to side), flushing the catheter with 10–20 mL 0.9% NaCl [43] or lumen reversal [15,42,44] may be potential solutions. This allows the reversible causes, such as a short-term obstruction of the catheter’s tip by the vein wall, to be eliminated [45]. 

Subsequently, if the catheter malfunction is preventing the hemodialysis procedure, thrombolytic therapy may be indicated [15,45]. A history of hypersensitivity to thrombolytic substances should be taken, and recent hemorrhage and severe thrombocytopenia must be excluded [44]. If no contraindications are found, a thrombolytic agent is administered into the lumen of the catheter in a volume depending on the type of CVC, usually between 1.9 and 3.1 mL. Administration of more than the volume recommended by the manufacturer must be avoided due to the potential adverse systemic effect. Unfortunately, no single effective protocol for thrombolysis has been developed so far. Depending on the hemodialysis unit, both the substance used, the method and the time of leaving it in the catheter’s lumen differ.

Currently, the most recommended thrombolytic therapy is to instill 1 mg of rtPA diluted with saline to the volume of the lumen into the CVC [42,46]. The higher 2 mg dose, suggested by KDOQI Guidelines, has equal efficacy in restoring the patency of the catheter in comparison with 1 mg rtPA, however, it is associated with higher costs [47]. Leaving alteplase in the lumen of the catheter for longer than an hour does not increase its effectiveness [48] due to its short, 4 to 6 min half-life.

Urokinase, widely used in the past, especially in Europe, was less effective compared to alteplase, especially in the case of complete catheter occlusion [46,49]. Its efficacy was better with the short, 30-min protocol than with the long, 48-h dwell time [50]. Currently, urokinase is not available in many European countries, including Poland.

In recent years, new thrombolytic agents have appeared, including reteplase. It has a different molecular structure than alteplase, which increases the half-life and improves penetration into the clot. Based on the studies conducted so far, it seems to be more effective than alteplase, however, there are no reliable, randomized clinical trials that would confirm these preliminary reports [45,51]. A thrombolytic agent with another mechanism of action, independent of the plasminogen activation system, is alfimeprase, a recombinant metalloproteinase. It binds to the α-chain of fibrin and directly breaks down the thrombotic material [45]. Despite initial, promising results in a small group of patients, this product did not enter widespread use. In Europe, alfimeprase is included in the list of orphan drugs only for the treatment of acute peripheral arterial obstruction.

Complications and adverse reactions of thrombolysis, while maintaining proper doses and volume of the administered agent, are extremely rare. No major bleeding was reported [42,45,47,51].

Effective thrombolysis allows for an immediate hemodialysis procedure, but rarely completely removes the cause of the catheter malfunction. Only 5–25% of cases, usually complete obliteration, are caused by an intraluminal clot [45] and only then thrombolysis is likely to be effective in the long-term. In most cases, catheter dysfunction results from the presence of a fibrin sleeve that grows along the foreign body towards its end over time and promotes the formation of a thrombus, which ultimately leads to impaired catheter function [45,52]. Unfortunately, the administration of a thrombolytic agent does not destroy the fibrin sheath, so it does not remove the main cause of CVC malfunction. For this reason, the effect of thrombolysis is short-lived, usually lasting 14–42 days depending on the study [48,52,53,54]. Thrombolysis should be considered rather as an emergency treatment aimed at providing the patient with the possibility of renal replacement therapy until elective catheter replacement is scheduled [48,52,53]. Some centers even estimate that the risk of bleeding and the costs associated with the thrombolysis are higher than the replacement of the catheter by a qualified operator and skip a thrombolytic therapy attempt [21,40]. Exemplary scheme of conduct in case of catheter malfunction is shown on Figure 3.

## 9. Conclusions

It is important to use the most effective prevention of complications, including thrombosis. Of the currently available substances, it seems that citrate at a concentration of 4% has the best cost-effectiveness and safety profile. This is reflected in the recommendations of the American Society of Diagnostic and Interventional Nephrology (ASDIN) and European Renal Best Practice (ERBP), which recognize 4% citrate and heparin as the recommended lock solution. Since the recommendations have been published, new studies have appeared suggesting the advantage of 2+1 protocols, i.e., the use of a standard lock solution twice a week and a thrombolytic agent or a combined solution containing it once a week. These strategies require confirmation in further studies, however, provided that the price is acceptable, they constitute an interesting direction in the development of CVC lock solutions. Regardless of the type of lock solution selected, if prophylaxis with a thrombolytic agent is chosen, it seems advisable to carry it out from the moment of catheter insertion throughout its service life to reduce the risk of thrombotic complications [31,55].

However, despite proper prophylaxis, each CVC will sooner or later become dysfunctional. Due to the pathomechanism, in case of a catheter malfunction, one should schedule elective catheter replacement, even if thrombolytic therapy restores catheter patency. Due to the many uncertainties resulting, among other things, from different definitions of catheter dysfunction and small sample size, further randomized trials in large populations of patients are urgently needed to determine the best possible prophylaxis and management of the hemodialysis catheter dysfunction.

## Figures and Tables

**Figure 1 jcm-10-02230-f001:**
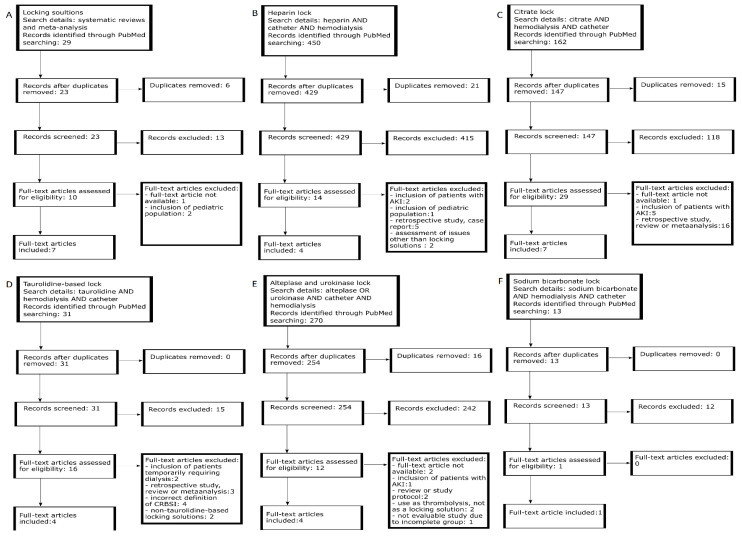
PRISMA flowcharts showing the search strategy and results: A—systematic reviews and meta-analysis; B—heparin lock; C—citrate lock; D—taurolidine-based lock; E—alteplase and urokinase lock; F—sodium bicarbonate lock. List of abbreviations used—AKI- Acute Kidney Injury.

**Figure 2 jcm-10-02230-f002:**
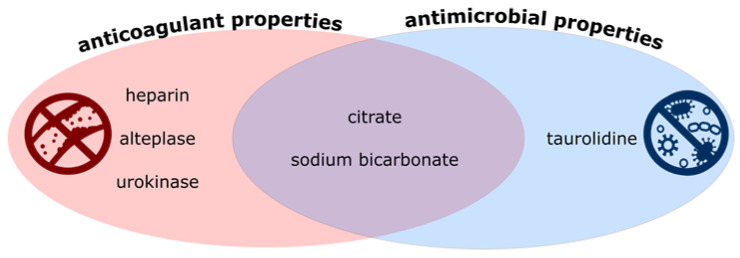
Summary of the properties of locking solutions. On the left, solutions with anticoagulant properties. On the right, locks with antimicrobial properties. In the middle, solutions with both of these properties.

**Figure 3 jcm-10-02230-f003:**
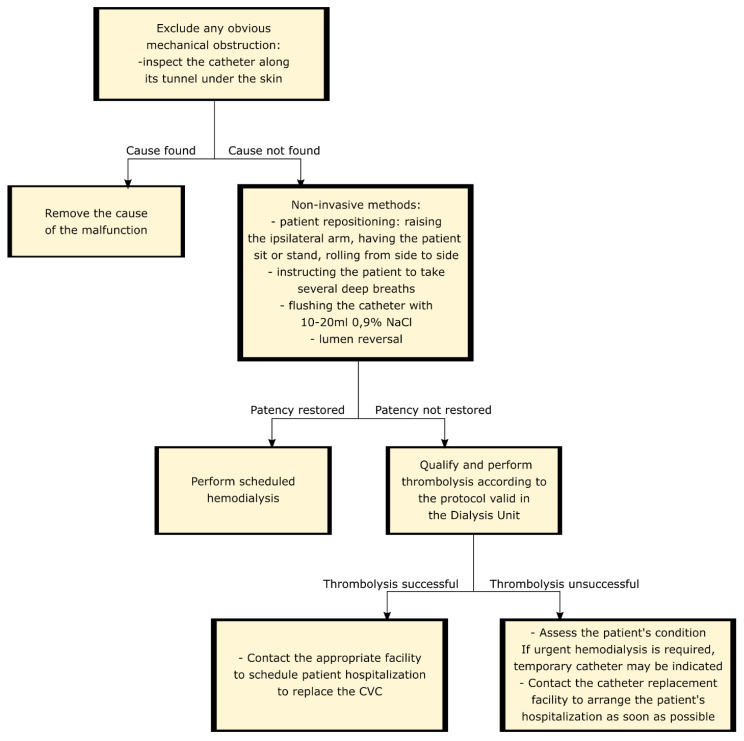
Exemplary scheme of conduct in case of catheter malfunction.

**Table 1 jcm-10-02230-t001:** Summary of key systematic reviews and meta-analyses.

Citation	Review Type/Reporting Quality	Description	Key Findings/Conclusions
Hemodialysis catheter locking solutions and the prevention of catheter dysfunction: a meta-analysis [6]	-meta-analysis -met 23 of 24 PRISMA items for meta-analysis	- both tunneled and nontunneled CVC included - both acute and chronic hemodialysis included - only catheter dysfunction assessment - locking solutions included: heparin, citrate (concentrations 2.2–46.7%); rtPA, ethanol, cefotaxime+heparin, tauroldine+citrate, hypertonic saline+heparin - included RCTs and observational studies published from 1980 to December 2013 - met PRISMA reporting criteria except for absence of protocol and registration	-no significant difference in catheter patency between heparin versus any other lock solutions -no statistically significant difference in catheter patency per 1000 days, catheter exchange or use of thrombolytic therapy when comparing locking solutions
Preventing haemodialysis catheter-related bacteraemia with an antimicrobial lock solution: a meta-analysis of prospective randomized trials [8]	-meta-analysis -met 23 of 24 PRISMA items for meta-analysis	- both tunneled and nontunneled CVC included - both acute and chronic hemodialysis included - only CRBSI assessment - locking solutions included: heparin 5000 U/mL vs. antimicrobial lock solution (ALS) (i.e.: gentamicin-citrate, gentamicin-heparin, taurolidine-citrate, minocycline-EDTA, citrate 30%, cefotaxime-heparin, cefazolin-gentamicin-heparin - included RCTs and observational studies published from 1999 to March 2007 - met PRISMA reporting criteria except for absence of protocol and registration	- use of an antimicrobial lock solution decreases the risk of CRBSI by approximately a factor 3. Although the under-representation of small studies with a non-significant or negative effect suggests an overestimation of the effect of ALS - no difference between tunneled and nontunneled CVC, possibly because of the under-representation of nontunneled CVC
A meta-analysis of hemodialysis catheter locking solutions in the prevention of catheter-related infection [9]	- meta-analysis -met 22 of 24 PRISMA items for meta-analysis	- both tunneled and nontunneled CVC included - both acute and chronic hemodialysis included - CRBSI and CRT assessment - locking solutions included: heparin 5000 U/mL or 1000 U/mL vs. antimicrobial lock solution (i.e.,: gentamicin, cefotaxime, minocycline, cefazolin, taurolidine-citrate, 30% citrate) - met PRISMA reporting criteria except for absence of protocol and registration, and assessment of the risk of bias in individual studies	- CRBSI was 7.72 times less likely when using ALS. Rates of catheter thrombosis did not increase - limitations are short duration of follow-up of the included studies and inclusion of both acute and chronic hemodialysis patients
Benefits and harms of citrate locking solutions for hemodialysis catheters: a systematic review and meta-analysis [10]	- meta-analysis -met 24 of 24 PRISMA items for meta-analysis	- both tunneled and nontunneled CVC included - both acute and chronic hemodialysis included - citrate locking solutions (not combined with other agents) vs heparin - all-cause mortality, bacteremia rates, all-cause hospitalization rates, access-related hospitalization, catheter replacement/exchange events, bleeding, and local/in situ thrombolysis assessment - included RCTs and observational studies published from inception to June 2013	- low overall quality of evidence due to heterogeneity and small sample sizes - significantly lower risk of bleeding in citrate group - rates of death and bacteremia tended to be lower with citrate but not statistically significant - no difference in hospitalization or patency-related outcomes
Anticoagulants and antiplatelet agents for preventing central venous haemodialysis catheter malfunction in patients with end-stage kidney disease [11]	- meta-analysis -met 24 of 24 PRISMA items for meta-analysis	- both tunneled and nontunneled CVC included - chronic hemodialysis - locking solutions included: heparin 5000 U/mL, low or no dose heparin, alternative anticoagulant locking solutions (i.e.,: citrates, antibiotic locking solutions, rt-PA, ethanol) - also systemic agents included - CVC malfunction, CRBSI, all-cause mortality, adverse events assessment - included RCTs published to January 2016	- among individual agents, only rt-PA reduced CVC malfunction (in 1 study) - low or no dose heparin may have the same efficacy in preventing catheter malfunction as 5000 U/mL - significant reduction on CRBSI for citrate, antibiotics, rt-PA, but not for ethanol or low dose heparin and systemic agents - additional use of antibiotic locks to citrate has no additional impact on CRBSI - no significant effect on all-cause mortality - reporting of safety outcomes was infrequent
Citrate versus heparin lock for hemodialysis catheters: a systematic review and meta-analysis of randomized controlled trials [12]	-meta-analysis -met 23 of 24 PRISMA items for meta-analysis	- both tunneled and nontunneled CVC included - both acute and chronic hemodialysis included - locking solutions compared: citrate locks vs heparin (5 studies); citrate+other antimicrobial solution vs heparin - CRBSI, exit-site infection, CRT assessment - included RCTs published to March 2013 - met PRISMA reporting criteria except for absence of protocol and registration	- citrate locks concentration 1–7% were associated with decreased CRBSI, whereas high concentrations (30–46.7%) had no effect - addition of antimicrobial substance to citrate was associated with decreased CRBSI - bleeding risk was lower for citrate than heparin - no difference in exit-site infection or CRT
Comparative efficacy and safety of lock solutions for the prevention of catheter-related complications including infectious and bleeding events in adult haemodialysis patients: a systematic review and network meta-analysis [13]	-meta-analysis -met 23 of 24 PRISMA items for meta-analysis	- both tunneled and nontunneled CVC included - both acute and chronic hemodialysis included - different locking solutions: heparin 5000 U/mL; low-dose heparin, antibiotics (e.g., cloxacillin, cefotaxime, linezolid, vancomycin, gentamicin) combined with anticoagulants (e.g., heparin, citrate, EDTA, urokinase) minocycline, taurolidine), ethanol - CRBSI, bleeding events, CVC malfunction, exit-site infection, all-cause mortality assessment - included RCTs published from the date of databases inception to August 2018 - met PRISMA reporting criteria except for absence results of individual studies	- ethanol and antibiotics combined with anticoagulant were more effective in preventing CRBSI compared to heparin - low-dose heparin and citrate had lower bleeding risk than heparin - using of gentamicin was connected to development of bacterial resistance after 6 months and to ototoxicity - no effect on CVC malfunction or all-cause mortality

List of systematic reviews and meta-analyzes referring to specific locking solutions described in the review—for heparin: reference 6; 8–13. for citrate: reference 6; 8–13 (in reference 8–9 only citrate in concentration of 30%). for taurolidine: 6; 8–9; 13. for alteplase and urokinase: 6; 11; 13. for sodium bicarbonate: no systematic review or meta-analysis was found. List of abbreviations used—ALS, antimicrobial lock solution; CRBSI, catheter related bloodstream infection; CRT, catheter related thrombosis; CVC, central venous catheter; EDTA, ethylenediaminetetraacetic acid; PRISMA, Preferred Reporting Items for Systematic Reviews and Meta-Analyses; RCT, randomised controlled trial; rtPA, recombinant tissue plasminogen activator. Systematic reviews and meta-analyses were assessed using the PRISMA checklist (for meta-analyses, the PRISMA checklist contained 24 required reporting items). Only the required items were used for scoring.

**Table 2 jcm-10-02230-t002:** Summary of the estimated cost of locking solutions.

Locking Solution	Vial Volume	Price in PLN/vial	Price in €
Heparin	25,000 IU/5 mL	15.04	3.3
Citrate 4%	5 mL	9.91	2.17
Citrate 30%	5 mL	7.64	1.67
Citrate 46.7%	5 mL	11.78	2.58
Taurolidine	6 mL	26.42	5.79
Taurolidine+heparine 500 IU	10 mL	40.82	8.95
Taurolidine+urokinase 25,000 IU	5 mL	146.72	32.15
Alteplase 10 mg	10 mg/10 mL	553.39	121.28
Urokinase	250,000 IU	627.41	137.5
NaHCO3	20 mL	3.40	0.75

Price per vial is given due to the possibility of using different concentrations and different CVC volumes. The price is given in PLN and converted into EUR based on the exchange rate given by Polish National Bank on 2021-05-10. Prices may vary depending on the center and the country. List of abbreviations used—CVC, central venous catheter; PLN—official code of the Polish currency; €—Euro.

## Data Availability

No new data were created or analyzed in this study. Data sharing is not applicable to this article.

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
