# Peer review of "Locked Away—Prophylaxis and Management of Catheter Related Thrombosis in Hemodialysis"

_jcm, 2021, doi:10.3390/jcm10112230_

Round 1

Reviewer 1 Report

Authors have accepted my previous recommendations and I suggest the revised manuscript is acceptable for publication.

Reviewer 2 Report

Authors have responded to all query and have  included them  in the  manuscript. Needs minor revision for the spell check and the it can be accepted for publication

Reviewer 3 Report

Thanks for update reads well.

This manuscript is a resubmission of an earlier submission. The following is a list of the peer review reports and author responses from that submission.

Round 1

Reviewer 1 Report

This is an excellent, comprehensive, well-written review article. It is suitable for publication in its current form.

A suggestion would be to add details of representative comparative costs of the agents discussed since this aspect of choice of agent is included in the report.

Reviewer 2 Report

The review is well conceptualized , focused content and good flow of the concept

1)        Focus on infection

2)        The most 48 common complications associated with central catheters include intra- or pericatheter   thrombosis (catheter related thrombosis - CRT) and catheter-related bloodstream infections (CRBSI)

Authors have cited 'Reference 4' , please provide the percentage of malfunction attributed to the thrombosis

3)        Table 1, with the following reference 8, 9 and 10 predominantly focuses on the CRBSI,  however this can be included and suggest to put critic in the discussion and the thormbosis and the CRBI would go along the spectrum f the complication of the instilling the locking solution

4)Authors have described the logistic of the search of  article  review for the locking  solution. Suggest i) take figure 1 before the table 1 in the flow of manuscript ii) in Table 1 put the studies for the other locking solution like Taurolidine iii) describe limitation of the systemic or multi-variate analysis for the

5)The conclusion if well described , but too length and confusing for the audience. Suggest keep it short describing only the current recommendation and what authors suggest should be the locking solution to prevent he thrombosis and the infection.

Reviewer 3 Report

Thanks for opportunity to review this.

I think it would be useful to include some sort of representative costs for the different locking products. These will vary between countries but if these were quoted for say Poland in Euros or US dollars I think it would be helpful.

The suggestion that such products should be used from the outset of line insertion seems sensible but is there evidence in the literature for all products that this actually makes a difference?  If not then recommendations should state this uncertainty.